# Integrating Mental Health Management into Empowerment Group Sessions for Out-of-School Adolescents in Kenyan Informal Settlements: A Process Paper

**DOI:** 10.3390/ijerph21020223

**Published:** 2024-02-14

**Authors:** Joan Mutahi, Beth Kangwana, Dorcas Khasowa, Irene Muthoni, Oliver Charo, Alfred Muli, Manasi Kumar

**Affiliations:** 1Department of Psychiatry, School of Medicine, College of Health Sciences, University of Nairobi, Nairobi P.O. Box 30197-00100, Kenya; joanmutahi20@gmail.com (J.M.); dkhasowa@gmail.com (D.K.); 2Population Council–Kenya-Avenue 5, 3rd Floor Rose Ave, Nairobi P.O. Box 17643-00500, Kenya; bkangwana@popcouncil.org (B.K.); alfredmuli22@gmail.com (A.M.); 3Integrated Education for Community Empowerment, Nairobi P.O. Box 7631-00300, Kenya; norshmiash3@gmail.com; 4Rapid Effective Participatory Action in Community Theatre Education and Development, Nakuru P.O. Box 15614-20100, Kenya; olivercharo14@gmail.com; 5Institute for Excellence in Health Equity, New York University Grossman School of Medicine, US-550 First Avenue New York, New York, NY 10016, USA

**Keywords:** out-of-school adolescents, Kenya, mental health, stigma, sexual and reproductive health and rights, task sharing, life skills, integrating, intervention

## Abstract

This article presents processes for developing contextualized training procedures to better appreciate partnership, capacity-building experiences, and specific implementation challenges and opportunities for mental and public health teams. The program enrolled 469 out-of-school adolescents to participate in the integration of youth mental health into health and life-skill safe spaces. The teams utilized various methods to achieve process outcomes of restructuring and adapting curricula, training youth mentors, and assessing their self-efficacy before integrating the intervention for 18 months. The Coronavirus (COVID-19) pandemic became an additional unique concern in the preliminary and the 18-month implementation period of the program. This necessitated innovation around hybrid training and asynchronous modalities as program teams navigated the two study locations for prompt training, supervision, evaluation, and feedback. In conclusion, out-of-school adolescents face a myriad of challenges, and a safe space program led by youth mentors can help promote mental health. Our study demonstrated how best this can be achieved. We point to lessons such as the importance of adapting the intervention and working cohesively in teams, building strong and trusting partnerships, learning how to carry out multidisciplinary dialogues, and continuous supervision and capacity building. This article aimed to document the processes around the design and implementation of this innovative intervention and present a summary of lessons learned.

## 1. Introduction

Adolescence is a crucial phase of neurocognitive, physiological, and social development, and is manifested by impulsive and risk-taking behavior and changes that lay the foundation for lifelong capacities and aspirations [1]. The developmental changes could be accompanied by adverse social determinants of health, such as an adolescent’s lack of understanding of their bodies, sexuality, identity, rights, economic insecurity, unequal gender norms, and pressure to conform with peers [2]. This is likely to increase potential future risks [1] of unprotected and early sexual debut, with related consequences such as unintended early pregnancies and exposure to sexually transmitted infections (STIs), including human immunodeficiency virus (HIV) [3], in addition to mental ill-health [4]. Mental health problems in adolescence leads to later life social maladjustment, reduced economic productivity [5], diminished physical fitness, and in some extremes a reduction in peak cognitive abilities [6,7]. As a result of greater social and economic deprivation, including poorer access to health care and other social services [8,9], adversities are significantly higher in urban informal settlements in low-income settings. Dropping out of school would therefore compound the likelihood of poorer mental health and financial outcomes, given the inability of adolescents to engage in meaningful academic pursuits, skill building, and peer and adult mentorship that often presents in schools [8]. 

Group-based empowerment sessions, built off a resilience-based model used to address the risks and unmet needs of adolescents in more marginalized settings who may not have access to formal social and health services such as schools and health care, could be effective [9]. This approach has been shown to improve an adolescent’s reproductive health and knowledge, livelihood outcomes, and gender attitudes and beliefs, and has been successfully used in a wide range of settings [10]. These sessions have mainly focused on topics covering sexual and reproductive health and rights (SRHR), health and life skills (HLS), and economic empowerment. However, a gap exists in evidence of how these programs can be leveraged to improve mental health (MH) outcomes [10]. The integration of MH interventions into SRHR, HLS, and economic-empowerment-building sessions for adolescents is likely to have multifaceted positive outcomes through reducing the exposure to adverse social determinants of mental illness, providing them with the skills to manage symptoms related to mental illness, and educating them on where to seek appropriate treatment [1]. 

Our study, Bridging the Gaps (BTG), was a unique attempt to evaluate the feasibility of integrating a mental health component into an evidence-based, mentor-led, social, health, and economic-empowerment program for urban, informal-settlement-dwelling, out-of-school adolescents in Kenya. The team anticipated the challenge of interrelating the dynamic factors affecting the identified population given the fact that they are Not in Employment, Education, or Training (NEET), and their possible emancipated status [11]. The Holistic Framework proposed by McElroy [12,13,14] guided the team in an attempt to successfully tackle barriers to improved well-being through its multidimensional approach while employing fidelity checks to ensure adequate integration of the adapted curricula. The program uses a task-sharing model that has been widely tested in sub-Saharan Africa (SSA), and has been promoted as a successful implementation strategy for scaling up mental health to improve outcomes in low- and middle-income countries (LMICs) [15,16,17,18]. The aim of this article is to document the processes around the design and implementation of this innovative intervention and present a summary of lessons learned. 

## 2. Materials and Methods

### 2.1. Study Setting

This study was carried out in two Kenyan urban informal settlements: Kariobangi in Nairobi County and Rhonda in Nakuru County. Kenyan urban informal settlements are typically characterized by high residential mobility, low-quality housing, high crime rates, and minimal government services [19,20]. Data from urban informal settlements in Kenya shows the following: 43% of girls 15–19 years old are not in school, mainly due to the inability to pay school fees; 13% of adolescent girls have symptoms of depression based on the Patient Health Questionnaire Version 9 (PHQ-9); 22% have experienced physical or sexual violence in the past year; and about 47% of girls have been exposed to adverse life events in the family in the past year [21,22]. These statistics have since been exacerbated as a result of the COVID-19 pandemic [21,23]. 

Furthermore, 15–18 is a crucial age to study as this is a transition point for adolescents who are out of school and moving into informal employment, training, and family life, among other endeavors. Additionally, school enrollment is generally high for younger adolescents in primary school. School dropout tends to take place as adolescents transition into secondary education and above. It is around this age when adolescents start to become sexually active and are more likely to be exposed to early pregnancies and sexually transmitted infections (STIs). Adolescents already experiencing these adversities are significantly more likely to experience mental illness and therefore we hope that this intervention will equip them with the skills to manage this challenging time.

### 2.2. Participants, Sites, and Design

The targeted sample size for this study was 400 adolescents equally split across the two sites. A total of 469 adolescents were enrolled for the baseline survey to account for potential loss to follow-up, while also keeping in mind the uncertainty brought about by the pandemic. Adolescents recruited into the study were out of school, between the ages of 15 and 18 years, residing within the two identified urban informal settlements, and did not demonstrate any severe mental disorder or disability that would prevent meaningful group participation. During recruitment, participants were screened for depression using the Patient Health Questionnaire-9 (PHQ-9) and for substance use using the World Health Organization (WHO) Alcohol, Smoking and Substance Involvement Screening Tool version 3 (ASSIST v.3). Those scoring high in severe mental distress, alcohol and substance use, and suicidal tendencies were referred to relevant psychiatric or psychological services.

More girls were selected compared to boys, because in these settings girls are generally more marginalized and therefore more likely to experience adverse outcomes such as sexual and gender-based violence (SGBV), early pregnancy, and financial exploitation, among others [24,25,26]. Additionally, the study team had a longer track record of carrying out mentorship sessions with girls and with this study were learning to integrate boys into more girl-centered programming.

Adolescents were recruited by trained mentors who resided in the areas. They carried out door-to-door recruitment for out-of-school adolescents in the two sites of interest. This was considered a better method of recruitment for the most vulnerable adolescents who may not be reached through more commonly used recruitment methods and avenues, such as through public advertisements, and therefore left out of program activities [22]. Mentors recruited study participants in two phases to reduce the amount of time carrying out door-to-door activities, which had an increased risk of exposure to COVID-19. In the first phase, mentors reached out to study participants through phone calls by obtaining numbers using the “snowballing” technique. Mentors originated from the study area and had access to the phone numbers of potential participants. In the second phase, if mentors were unable to reach the target number of participants through phone calls then they had to go into the community for door-to-door recruitment. Mentors had been trained in community-based door-to-door recruitment of out-of-school adolescents into safe space groups.

By the end of the study, 18 months, a total of 339 out-of-school adolescents remained in the study with the rest lost to follow-up [See Figure 1, Bridging the Gaps CONSORT flow diagram, below].

### 2.3. Teams Involved in Bridging the Gaps Feasibility Study

#### 2.3.1. The Implementers—Community-Led Organizations

Two local organizations were involved in the study implementation as they had previous experience leading community interventions through youth mentorship and mentor-led groups in various programs. As such, they took part in the selection and recruitment of community lay youth mentors residing within the areas. The organizations were also significantly involved in the referral mapping to be used when mental-health distress, or other health, legal, financial, and social-related issues, were identified in any of the participants. This allowed for prompt referrals to relevant services from stakeholders within the regions. Such stakeholders included police-gender desks in SGBV cases, local financial and training institutions, nonprofit organizations, community health care workers, psychologists, and psychiatrists, among others. The two organizations involved are detailed below.

At the Nairobi Site, the Integrated Education for Community Empowerment (IECE) is located in Kariobangi. This is a nongovernmental organization (NGO) present in Nairobi and other Kenyan counties. It focuses on adolescents, women, and children living in informal urban and rural areas. It aims at empowerment [27] through the provision of basic and low-intensity counseling support using structured participatory approaches used in different projects and initiatives.

At the Nakuru Site, the Rapid Effective Participatory Action in Community Theatre Education and Development (REPACTED) is located in Rhonda. REPACTED is a group comprised of trained youth volunteers. Their mission is to promote behavior change and communication in the shaping of experiences, strength identification, challenge tackling, and success celebration of a multiculturally diverse network of adolescents and youth within Nakuru county, Kenya [28].

#### 2.3.2. Youth Mentors

The local organizations recommended selected lay members residing within the study areas to be youth mentors. Most of the mentors had been part of a similar project titled NiSikilize Tujenge Pamoja (NISITU), loosely translated to mean “*listen to me so that we build together*” [29,30,31], a study led by Population Council-Kenya (PC-K) that engaged men and boys in girl-centered programming. Female mentors between 22 and 35 years old, from the communities of implementation, and with prior experience in facilitation, working with adolescents and sexual and reproductive health were selected. One critical component of the screening was to assess their attitudes on adolescent sexuality/use of contraceptives, mental health and gender norms. Male facilitators selected were between 25 and 40 years old, from the communities of implementation, and with prior experience in facilitation, work with adolescents and sexual and reproductive health. One critical component of the screening is to assess their current attitudes toward mental health, gender norms and openness to gender transformation. The teams did not accept mentors without “progressive” attitudes in these areas. The mentors underwent a skills- and capacity-building exercise at the start of the study.

#### 2.3.3. Research Team

A team of researchers from PC-Kenya was responsible for designing and implementing the research framework around the program. This included the baseline and endline quantitative data collection activities, the formative and midline qualitative activities, and the routine monitoring of the program. PC-Kenya has had several years of experience designing and implementing youth mentor-led group sessions focused on improving outcomes in sexual and reproductive health, life skills, and gender norms, mainly targeting adolescent girls [30,31,32].

#### 2.3.4. Mental Health Team

The mental health team was composed of a lead mental health researcher who was also an adolescent expert and clinical psychologist, and two postgraduate clinical psychology students. The lead researcher provided oversight on the mental health intervention design and collaborated with the students on mental health curriculum development, mentor training, supervision, and participant referral.

### 2.4. Phases of the Study

The various teams came together to work on this study during the COVID-19 pandemic, which called for significant adaptability in effectively responding to challenges and navigating the ongoing pandemic situation. The phases are discussed below as guided by Figure 2 from Section 2.4.1 to Section 2.4.9.

#### 2.4.1. Formative Research

Before commencing the study, formative research was carried out, aiming at qualitatively understanding the knowledge, attitudes, and access to care for out-of-school adolescents with mental illness residing in urban informal settlements. Forty-two virtual in-depth interviews (IDIs) were conducted with out-of-school adolescent participants, their parents, and mentors from each site. The interviews were recorded with participant consent, transcribed, and then coded by two independent qualitative analysts using Atlas.ti Version 8 software into themes and subthemes. A select number of the transcripts were also double-coded to ensure common understanding and interpretation, and an acceptable inter-coder agreement was also reached for reliability (coefficient of 0.70). The findings of this exercise were reviewed and informed the methods of the feasibility study. The interview guide and results of this formative research have been included as Appendix A.

#### 2.4.2. Formation of an External Advisory Group and Consultation

This study began with a meeting of a select group of 12 persons working in the private, public, and NGO sectors who provided intellectual input into the study pertaining to its design and findings, formally referred to as the BTG-External Advisory Committee (EAC). The individuals had also been selected based on their experience working in mental health and/or adolescent health in their professions. This, and other subsequent meetings with the BTG-EAC, helped to ensure that this study was effective and relevant to the two communities and beyond.

#### 2.4.3. Curriculum Adaptation and Integration

##### Health and Life Skills Curriculum

The initial curriculum development of the core component of the girls’ curriculum, the Adolescent Girls’ Initiative-Kenya curriculum [25], was completed at the preliminary stages of this study. It was slightly adapted to factor in the study participants’ age and context. This was followed by the development of the male curriculum pulled from Tuko Pamoja’s *“we are together*”, a curriculum also developed by PC-Kenya, and the Program for Appropriate Technology in Health (PATH) [32], adapted from the Engaging Men as Partners Curriculum [33]. These two curricula, used in a previous study, NISITU, were incorporated. The girls’ curriculum was aimed at empowering participants in their engagement with boys and men through understanding personal attitudes and beliefs surrounding the different topics of health, life skills, finances, and gender norms. This content was then adapted for boys to allow them to gain skills and information on how to change attitudes, beliefs, and behaviors and become allies of girls and women in their lives. The content was novel and, therefore, the teams were focused on assigning the topics based on the already-existing locally contextualized social norms [see Figure 3]. Moreover, the curriculum gave room for joint sessions at various time points. The aim was to discuss and hear from the other sex about their thoughts and opinions about relevant topics previously discussed in the single-sex groups. 

##### Mental Health Curriculum

The development of the mental health curriculum was based on the original WHO-recommended Problem Management Plus (PM+), which is a five-session, individual, low-intensity psychological intervention program [see Figure 4 below]. This program is tailored to be delivered by nonspecialist counselors to address common mental disorders in adults affected by adversity [34,35,36,37]. The team adapted this intervention into one that would be offered in group and individual sessions for young people. The PM+ sessions focused on problem-solving, relationship strengthening, stress management, mental health first aid, and behavioral activation strategies. These topics were not only to be taught as separate sessions but also integrated into the other health and life-skills topics.

The PM+ group sessions were aimed at mental well-being promotion while the individualized sessions were aimed at treatment of mental health [Figure 4 describes the PM+ curriculum]. The training of mentors helped them identify at-risk adolescents in their mentor-led groups. For instance, pregnant adolescents, young mothers, and substance users would be identified, then the mentors would reach out and provide individual psychological help. This would focus on the adolescent learning how to address emotional situations and interpersonal problems, stress management, and behavioral activation to curb inactivity cycles common in depression while establishing active social support networks to help them live well.

Adaptation of the intervention was carried out in consideration of our target population who were lay youth mentors. The mental health team adapted the curriculum through multiple virtual weekly meetings reading through the WHO PM+ manual and various consultations with PC-Kenya and the implementers to understand the population targeted. The stakeholder meetings were aimed at content consultation to ensure the acceptability and ease of use of the manual.

Specific areas of the PM+, including but not limited to utilizing known relaxation techniques, strengthening already existing social engagements as support systems, and available behavior activation strategies, were adapted to fit into the culture and context of the target mentors and out-of-school adolescent population. These were considered in conjunction with suggestions from the implementing partners who were familiar with what the adolescents would find relatable. During the training, mentors also suggested other strategies, such as guided imagery of commonly available pleasurable items and experiences to be applicable to the Stress Management relaxation activities in Session 1, football club participation for the Behaviour Activation in Session 3, and beauty training for strengthening social support in Session 4, among others. It was imperative to remain appropriate and relatable for the intervention to be effective in the long term.

#### 2.4.4. Referral Pathway Mapping

Referral pathway mapping exercises were conducted early in anticipation of sustainable, timely, and more technical intervention in areas that could only be solved through formal services beyond this study. This was in collaboration with the mentors who had more experience in program implementation within the two communities. Recommendations from the mentors assisted in facilitating more targeted help for adolescents. They were able to identify referral needs through the PM+ assessments, general safe group conversation, evaluation of those lost to follow-up, and the adolescents approaching them and sharing their stories. Referral needs ranged from severe mental health issues, need for urgent financial help, exposure to SGBV, facing legal issues surrounding getting justice for reported SGBV cases, being in emergency situations of adolescent pregnancy, and needing interventions to prevent HIV and STI infection, among others.

Anticipating that there would be some participants presenting with more extreme psychological distress and related severe life difficulties, it was imperative that the mentors refer them to Primary Health Care facilities (PHCs) in the community for more targeted triage and support. The mental health team, through its partnership with other agencies such as the Ministry of Health (MoH), had previously trained a number of healthcare workers in Kariobangi on mental-health interventions allowing for referral support in psychiatric facilities and clinics within Nairobi county. Some referrals were identified in both Nairobi and Nakuru, including psychological counseling for suicidal adolescents, legal and rescue home support following SGBV reporting, as well as short-term skill training including in beauty and hairdressing for adolescents who expressed a need and interest. A sheet of referrals and assessments was maintained from the start of this study and the mentors would keep track of those who had been referred for more intensive support in the later stages of the intervention.

#### 2.4.5. Ethical Research Conduct and Child-Safeguarding Procedures

The research staff, including mentors, were trained in ethical conduct in participatory research. Mentors were sensitized to delivering psychotherapeutic interventions for young people. An additional Child Safeguarding training was conducted by external expert trainers in youth populations, and key standard operating procedures were developed as part of this work.

#### 2.4.6. Integrated Curriculum Training and Skills-Building for Youth Mentors

An intensive 15-day training schedule in the various curricula was organized to guide training [See Figure 5]. 

The overall objective of the training was to build the capacity of mentors in facilitating safe spaces groups and effectively delivering health, life skills, financial education, gender norms, and mental-health curricula to adolescent girls/boys in the program. The specific objectives of the training were to enable mentors to fully comprehend the concepts of the Bridging the Gaps feasibility study, acquire accurate information on the curricula, and practical skills to facilitate a range of interactive methodologies to be used during the safe space sessions.

Youth mentors were taken through the objectives of the BTG study, the theory of change, and different program components. The training also comprised a review of the basic tenets of mentorship and an evaluation of the experiences of the mentors previously involved in the NISITU program. The implementers trained youth mentors on team science, stakeholder engagement, mentor attitudes, group facilitation qualities, and the engagement and identification of adolescents’ needs as part of the preliminary activities of the training. Feedback was taken during program implementation considering that mentor selection was based on previous experience implementing the NISITU program. Reports indicate that mentors experienced increased self-efficacy in their ability to facilitate the mentor-led groups on the core curricula throughout this study.

Mentors needed to attend the training with at least 80% attendance. They also had to show a significant improvement score on the Mental Health and Literacy Score (MHLS) tool administered before and after the training. They needed to submit examples of group role-plays at the end of the training to demonstrate their grasp of the overall curricula. It was emphasized throughout that as the teams were testing a task-shifted model, their purpose was to be facilitative and supportive in the learning of the content. Role-plays ensured that the mentors applied the PM+ skills and were given prompt feedback. They presented contextual and relevant case examples mostly from scenarios that they themselves experienced as adolescents. The next phase of the study was rolled out when all youth mentors attained the basic requirements and qualifications of the criteria discussed. More information on this phase is discussed in the Process Results Section of this article.

#### 2.4.7. Mental Health Curriculum Refresher Training Mid-Intervention

Based on mentor feedback and an online researcher-developed questionnaire that identified gaps in competencies, two mental-health refreshers were set up. These refreshers were needed because this was the first time that the mentors were handling such a component in the mentor-led groups. Further, the mentor-led group sessions focused on the HLS curricula in the first few months of the implementation phase. Thus, a three-hour online skill transfer refresher session was conducted halfway through the program to boost the fidelity to the package followed by another refresher training day six months later at each site.

#### 2.4.8. Integrated Intervention Implementation

Upon a satisfactory level of mentor preparedness and self-reported willingness, the delivery of the integrated intervention into the already-existing safe space model commenced. It was established that this intervention would take place over 18 months. Implementation involved an initial facilitation of mentor-led group sessions focusing on content in financial literacy, gender norms, health and life skills, and mental health, followed by an integrated delivery of the same. Separate safe space group sessions for girls and boys were held once a week for a series of facilitated sessions. Five months into the intervention, joint monthly meetings between the boys’ and girls’ groups aiming at sensitizing participants on appropriate gender-sensitive behavior were held and co-facilitated by their respective female and male youth mentors. Debrief discussions on what was learned would take place in their regular groups afterward.

Additionally, group sessions were implemented during the COVID-19 pandemic, which meant that set rules and restrictions were followed throughout. During the intervention, teams limited the number of adolescents in a group to ten, provided masks and hand sanitizers, and saw to it that physical distancing was practiced. At some point during the study, the teams explored the practicality of having the sessions on the phone, which was reconsidered following the responses given by the mentors and mentees on a researcher-generated questionnaire citing concerns such as lack of access to mobile phones and the internet.

##### Weekly BTG Implementation Update Meetings

The BTG implementation team attended weekly implementation meetings from the start of the study where agendas were shared beforehand and action points assigned to various groups after each meeting. The overall aim of the meetings was to monitor the implementation of the program and troubleshoot any challenges promptly so as to maintain high-quality work throughout. Moreover, monitoring data collected by mentors were continually displayed on the Microsoft Power Business Intelligence (PowerBI, version 2.124.581.0) dashboard and used to support discussions on topics addressed, attendance, and number of sessions held, among other aspects of the study. Also discussed in these meetings were curriculum, training schedules, mentor-led groups, supervision, successes, and challenges

Furthermore, during implementation, a meeting between the BTG implementation team and the BTG-EAC was organized and held in September 2021. Its focus was on giving a report on progress and highlighting challenges and lessons learned to a party of external experts who would then discuss how to improve the feasibility study.

#### 2.4.9. Supervision

Supervision focused on the mental health component of the intervention since this was a new component for the mentors, ensuring that the quality and fidelity of the PM+ were maintained. The mental health team kept track of mentor skills, provided guidance on complex cases raised for management, gave timely feedback, assessed adolescents’ improvement in the course of the intervention, and sustained mentorship for proper adaptation of the curriculum. Innovative approaches to supervision, including WhatsApp group creation and real-time communication with mentors, listening to group recordings, and the physical observation of sessions were considered and implemented given the COVID-19 context, as discussed in Figure 6.

#### 2.4.10. Monitoring, Feedback, and Evaluation

Real-time monitoring data were collected throughout the intervention period. Regular monitoring allowed for program staff to track the progress of the different sessions and provide immediate support to groups that were displaying lower attendance rates. Mentors were trained to enter session details onto digital tools that had been developed on the web-based platform Open-Data-Kit (ODK) and upload them onto the server at the end of each session. A data manager visualized the data using Microsoft PowerBI and shared this with the program staff for review and action. Data were collected regarding the number of participants, topics discussed in the group sessions, one-on-one psychosocial help sessions, and community sensitization activities. Mentors were also allowed to enter other information regarding each session that they felt was important to highlight. Data collected on the one-on-one sessions also included details on any referrals made. These approaches enabled the teams to analyze and share insights on the implementation phase progress and clearly identify challenges being experienced while still ensuring that data accuracy and security were maintained. Thus strategic decision-making was experienced throughout the implementation phase.

## 3. Process Results Section

### 3.1. Curricula Restructuring

The curricula adaptation phase took a total of five months for all teams. This included an intensive review of the original manuals and regular team meetings with a view to fully capturing the needs of the target population. A brief breakdown of the adaptation is shown in Table 1.

### 3.2. Training Outcomes

#### 3.2.1. Youth Mentors

The age of the youth mentors recruited to facilitate the adapted curricula ranged from 24 to 35 years with the average age being 29, and all residing within the study locations. There were 16 female and 6 male mentors. Site-specific characteristics included the fact that in Rhonda, 70% had more than one year of mentoring experience. A minimum of two to seven years of experience was reported. For Kariobangi, 83% had been mentoring for more than one year, with the range being between three and seven years of experience. Pre- and post-training assessments were carried out on youth mentors. There was a total of 22 youth mentors at pre-training, with 12 from Kariobangi and 10 from Rhonda, which reduced to 19 by the end of the training where 9 remained in Kariobangi and 10 in Rhonda. The dropout was attributed to the mentors having competing engagements as well and some were present only as a backup in case other mentors dropped out mid-intervention. The teams needed to consolidate their knowledge of basic mental health which was evaluated using the Mental Health Literacy Scale (MHLS). This was because their mentoring experience was not entirely related to mental-health content facilitation. The pre-training scores ranged from 90.2 (SD:9.2; ranging from 77 to 107) to 96.1 (SD:6.1; ranging from 88 to 106). By the end of the training, the mean score had increased to 104 (SD:9.9) in Kariobangi and 106 (SD:7.6) in Rhonda.

The mentor’s feelings around integrating the mental-health package in the already-existing mentor-led groups were that addressing the other concerns in the financial, HLS, and gender norms curricula would help the adolescents deal with the practical sources of their emotional adversity; the other packages offered more “practical” solutions than simply talking through issues in a mental-health intervention. By the end of the training, mentors expressed an understanding that their role would be to empower adolescents to manage their own adversity in a way that best applies to their current situations regardless of whether the problems are of a financial, HLS, or mental health nature. Post implementation, mentors reported in a qualitative survey that they were adequately prepared for the intervention as is discussed in the team’s qualitative manuscript in preparation at the time of writing this article.

The mid-intervention informal assessments presented mentor efficacy on the continued facilitation of the PM+ mental-health curriculum. They also informed the decision to have the one-day refresher training, also discussed above, that focused on how to deal with adolescents, building on competencies, and identifying mental-health facilitation needs. Mentors who reported low self-efficacy were given personalized attention during the refresher and supervisory sessions to address their main concerns and improve their feelings about their ability to continue with the intervention. The survey had seven questions and the results are as shown in Figure 7.

#### 3.2.2. Training Schedule and Modalities

The training took up a hybrid mode due to the COVID-19 restrictions at both sites and took a total of 15 full days in early 2021 from January to February. The Public and Mental Health teams carried out a concurrent type of training. This means that the Reproductive Health, Life Skills, Financial Education, Gender, and Gender Norms curricula were provided at one site, and the other site received the Mental Health PM+ curriculum. This was considered innovative as one site was in the capital of the country, Nairobi, and the other was in a different county, Nakuru. The youth mentors were able to meet the study teams and understand the objective of the study [see Figure 8 on training meeting photos and program notes]. They attended full-day sessions that were interactive and full of role-plays to practice content delivery. Table 2 describes a brief overview of the division of the training phase.

### 3.3. Study Intervention Integration

#### 3.3.1. Supervision Outcomes

The supervision process yielded results on referrals made as well as safe space progress [Table 3 presents an example of a PM+-specific supervision report]. The evaluation ranged from the weekly team meetings discussing the real-time monitoring data on PowerBI to physical one-on-one and group mentorship sessions for mentors offered by the clinical psychologists. Findings showed that the mentors were able to hold the safe space sessions and make referrals to the relevant authorities and services upon identifying needs. The information gathered during midline (July–August 2021) helped troubleshoot and support the mentors to better continue with the implementation. For instance, a team-building session was organized for all teams as burn-out was experienced by the mentors. These challenges with compassion fatigue were encountered and exacerbated by the mentors’ own personal challenges faced due to the global COVID-19 pandemic.

#### 3.3.2. Participant Retention Outcomes

Given the continued supervision and monitoring conducted by the teams implementing the program, significant improvement in participant retention for the curricula facilitation was noted soon afterward and an example of the PowerBI dashboard representation is shown in Figure 9.

## 4. Discussion

There is limited information on how mental health integration in interventions can be achieved to target marginalized out-of-school adolescents. This article, just like similar reviews in sub-Saharan Africa (SSA), takes steps to fill this gap in knowledge by discussing the process and lessons learned in delivering such an integrated approach [38,39]. The WHO PM+ curriculum has been adapted recently in other SAA contexts, and evidence suggests that it is appropriate, acceptable, and feasible for multiple modalities, even telephonic delivery [40]. This is similar to what we conclude in this reflection. Coleman and colleagues [41], in a multisite adaptation in Rwanda and Malawi and two middle-income countries, Peru and Mexico, discuss the need to adapt the PM+ for local health systems for routine care. Despite the need to adapt diverse integrated mental health interventions in the region [16], there is a gap in studying their feasibility [4], making this study relevant. There is an estimated 85% treatment gap for mental problems in SSA [42] that has escalated with the COVID-19 pandemic creating disparities in access to support both formally and informally [43,44].

While this intervention persuades us to conduct further investigations among out-of-school adolescents, we have secondary information from SSA indicating that the challenges faced in establishing consistent opportunities for these groups of adolescents to meet and prioritize mental-health discussions are important and should be integrated with livelihood support [39,40]. Out-of-school adolescents do face challenges on various fronts and a Safe Space program led by youth mentors can try to resolve them. However, intervention and implementation science would work best if a multipronged approach of stakeholder engagement were able to offer lived experiences to the teamwork in the navigation of various challenges in the course of intervening [39,40]. The role of guided supervision during this intervention further identifies a new way of ensuring that quality checks and efficiency are maintained and linkage to psychological or specialized care is provided as a consideration of what works and can be adopted.

Various challenges made task sharing complicated. For instance, the process of understanding different stakeholders’ needs, selecting people, and training was protracted. There was a need for different types of funding prioritization to increase support for specialist referrals. As with most community-based interventions, there were mixed community expectations on the levels of engagement. However, the teams aimed to mitigate and resolve the aforementioned barriers and challenges in the course of implementation through teamwork and communication within and between PC-Kenya’s mental health team and the implementers.

The adolescents presented with a variety of lived experiences and a blend of competencies. Thus, one of the lessons learned in the course of the study was that it was not possible to ensure that all participants were exposed to the full intervention since they did not always attend all the safe space sessions. Challenges encountered needed solutions where, for example, during Mental Health curriculum supervision, it was noted that there was a need for continued reevaluation to resolve incomplete sessions. Mentors were encouraged to reach out to those who dropped out and see how to reengage. It was also required that they reevaluate specific participants for one-on-one sessions and refer others for psychological evaluation. Moreover, they had to follow up on the uptake for those who had missed opportunities to make the referrals.

The COVID-19 pandemic also created extreme difficulties in making mentor-led groups possible. Lessons were learned on how the pandemic made it difficult for more physical check-ins to be conducted which would have improved participant retention and stakeholder connectivity. However, conducting multifaceted interventions with limited face time may not be ideal in some study processes and troubleshooting scenarios in physical meetings increases the likelihood of mutual understanding and interpretation of challenges.

The implementers recommended that mentor selection be based on previous experiences implementing the adolescent safe space program NISITU. Issues surrounding participant retention were identified as exacerbating diminishing mentor morale and overall confidence. Youth mentor ages ranged from 22 to 35, which, despite the need to leverage past experiences facilitating mentor-led groups, might have been a limiting factor for this study.

Another lesson learned in this study was the need to clearly define the translation process and to ensure that common adolescent-favored language variations and dialects, like ‘*sheng*’, local slang commonly used in Kenya, be adapted in all materials produced from the assessment tools to the manuals. This could ease mentor hassle in facilitating the content in the mentor-led groups as well as increase a sense of ownership of the entire process by the adolescents in future research.

Midway through the program, burnout among youth mentors was observed. This necessitated a combined refresher and team-building activity. Burnout could have been a result of the PM+ being a more intensive component of the study and because it was also the first time that the mentors were facilitating mental-health-integrated mentor-led groups. Furthermore, the difference in mentor confidence during the previously handled core mentor-led groups’ curricula in the NISITU program and the mental health package was evident. It was therefore understandable that mentors would need continued capacity- and confidence-building exercises in the life course of the study. To curb retention challenges, it is crucial to budget for regular incentives and rewards for those who attend a minimum number of sessions. This, in most instances, would acknowledge the needs of the targeted population given that motivation is required for them to learn the skills taught in safe spaces. Moreover, some refined inclusion criteria for mentors could be required to ensure youth-friendly task sharing in implementation that keeps adolescents eager to interact with peers.

The authors wish to highlight that the study was instrumental in pointing out training and implementation needs facilitating an avenue to be innovative. The team optimized resources to achieve a digitally motivated prevention and treatment intervention in the heat of the COVID-19 pandemic. For instance, when it comes to training on integrated curricula, this study informs that adding an additional component to the curricula demonstrating how to integrate the curricula in the interventions should be considered in future research. This could include having specific case study discussions on how mentors need to make clear linkages between the core and mental health packages. Furthermore, incorporating training that is spaced out and accommodates the intensity of the mental-health curriculum, as well as a timely planned refresher period and regular check-ins on what is going on and how the mentors feel, could boost confidence and minimize the need for multiple refreshers.

Overall, the team believes that understanding the resources within the youth mentor group, breaking down the relevance of the intervention, and conducting repeat refreshers and continuous supervision proved to be highly motivating and beneficial. Moreover, the team is proud of its ability to mitigate participant recruitment and retention challenges through various means, including recruiting participants after the festive season, conducting door-to-door recruitment activities, and providing incentives such as diapers and flour left over from other projects run by the implementers to help in meeting some of their needs. This study also included a Safe Space Champions System where mentors identified those who continually showed up to help remind and reach out to other mentees in the case of absence. Considering the COVID-19 pandemic context, it was novel and showed innovation as the team came up with creative ways of connecting, such as the frequent WhatsApp chats with the mentors for supervision and Zoom sessions amongst team members for timely monitoring and evaluation. This ensured that the study continued within its stipulated timelines while ensuring quality management.

## 5. Conclusions

This article presents reflections on capacity building and training of lay health workers or youth mentors in two sites in Kenya to carry out multidisciplinary work targeting at-risk adolescents where mental health is a key aspect. We point to lessons such as the importance of adapting the intervention, the importance of working cohesively in multidisciplinary teams, building strong and trusting partnerships and multidisciplinary dialogues, as well as continuous supervision and capacity building.

## Figures and Tables

**Figure 1 ijerph-21-00223-f001:**
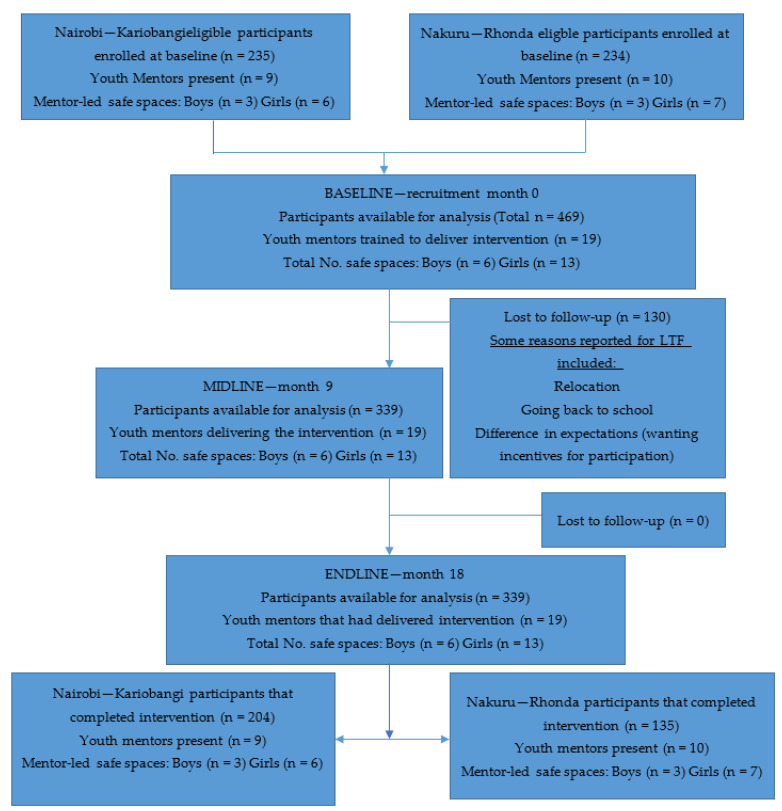
Bridging the Gaps CONSORT flow diagram.

**Figure 2 ijerph-21-00223-f002:**
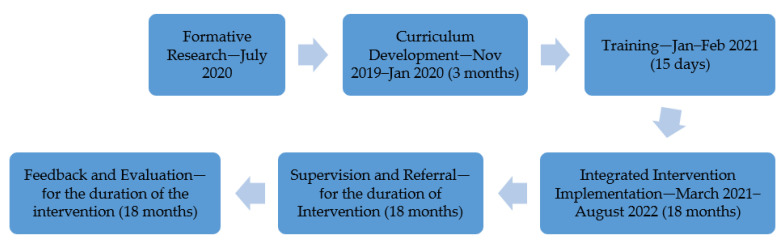
Phases of the study.

**Figure 3 ijerph-21-00223-f003:**
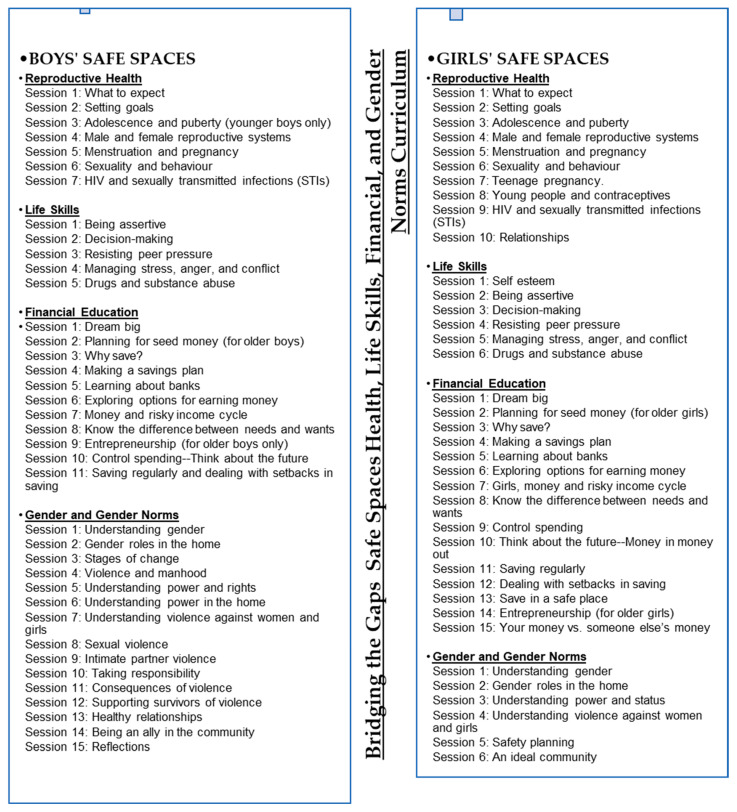
Bridging the Gaps (BTG) safe space health, life, financial, and gender norms curricula adaptation session outline.

**Figure 4 ijerph-21-00223-f004:**
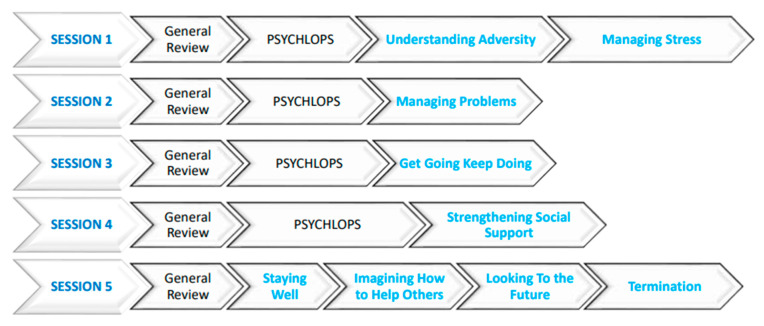
Mental health WHO PM+ adaptation curriculum session outline.

**Figure 5 ijerph-21-00223-f005:**
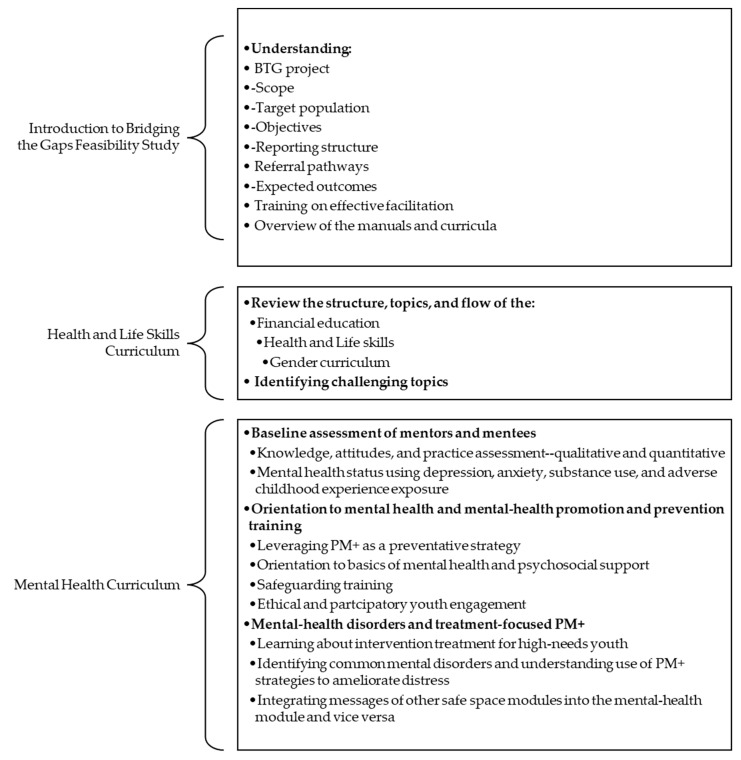
Bridging the Gap feasibility study training phase.

**Figure 6 ijerph-21-00223-f006:**
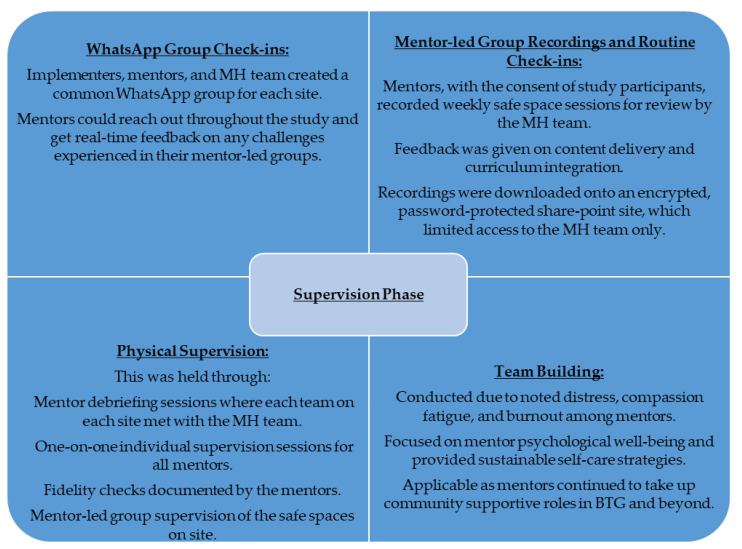
Supervision Phase.

**Figure 7 ijerph-21-00223-f007:**
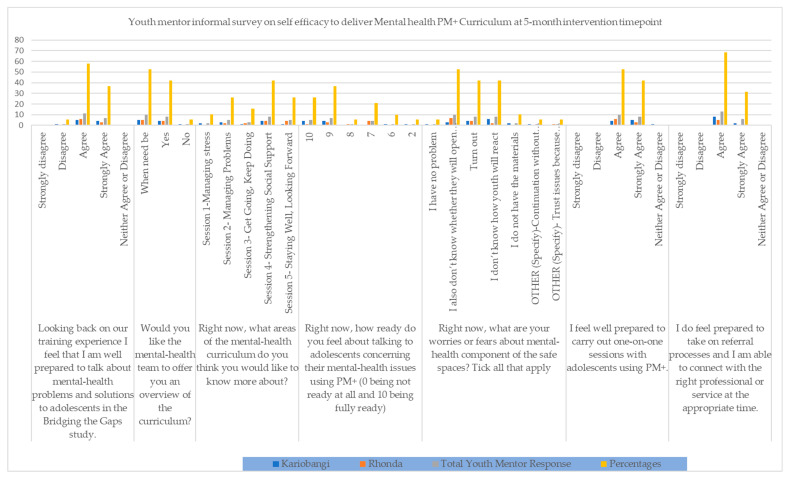
Youth Mentor Informal Survey on Self Efficacy to deliver Mental Health PM+ Curriculum at 5-month Intervention mark (May 2021).

**Figure 8 ijerph-21-00223-f008:**
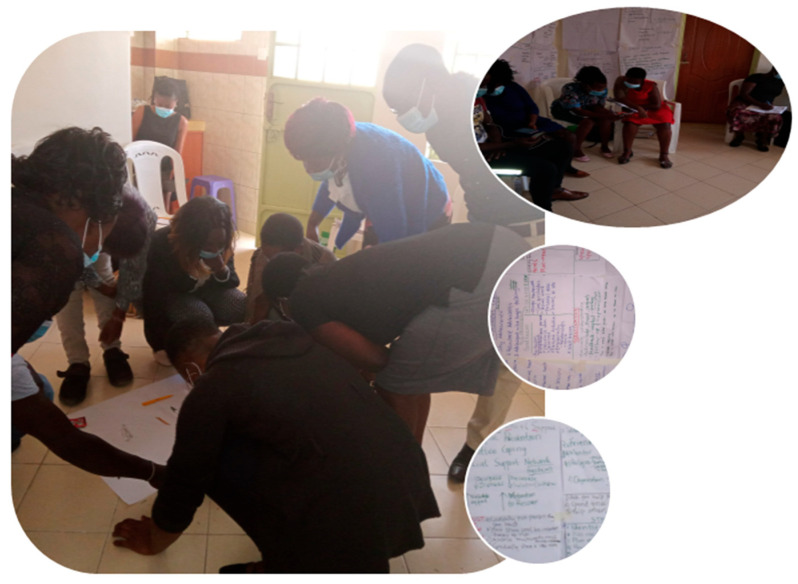
Images from the Youth Mentors’ Curricula Training Sessions–Rhonda.

**Figure 9 ijerph-21-00223-f009:**
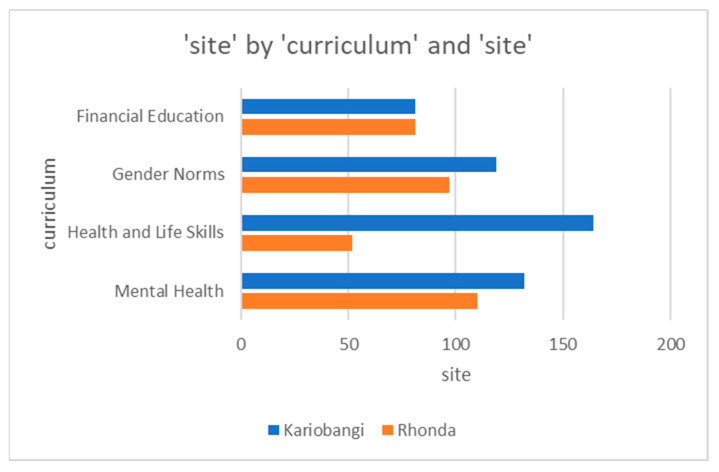
PowerBI dashboard representation of comparisons across modular curriculum delivery across sites.

**Table 1 ijerph-21-00223-t001:** Bridging the Gaps Curricula Adaptation and Restructuring.

Curriculum	Original Aspects	Adaptation/Restructuring Made
Adolescent Girls’ Initiative- Kenya curriculum	Designed for girls aged 11–14 with some topics suitable for only those on the older spectrum Has 4 different packages 3 packages aiming at a combination of multicomponent, multilevel interventions, and 1 focused on a community-level intervention Administered in weekly (1–2 h) group settings led by mentors Provides necessary referral based on need Some sessions allow for guest speakers No payment for participant attendance–voluntary Designed for those in school and out of school Facilitation through discussions, brainstorming, educational games, and role-plays with energizers as needed Has 28 sessions	Divided into 37 sessions
Tuko Pamoja curriculum (We Are Together) for Boys and Men	Adolescent reproductive health and life-skills curriculum Incorporates peer mentorship Sessions are divided into different topics that can be handled at different times Administered in weekly group settings	Divided into 38 sessions
NISITU curriculum–Engaging Men as Partners	Combination of the first 2 curricula (AGI-K and Tuko Pamoja) An asset-building approach to expand opportunities and reduce vulnerabilities (increase their power in relationships) for adolescent girls Applies the socio-ecological framework Using the theory of change, it aims at engaging boys and men in an evaluation of their attitudes on gender roles and power in relationships with girls and women Empowering boys and men increases their health and financial assets and will help achieve more equitable gender roles and reduce violence against girls and women To be delivered in same-sex safe spaces sessions Have monthly mixed sessions for both adolescent girls and boys Covers: Sexual and reproductive health skills, life skills, gender norms and power relations, and financial education	
WHO PM+	Low-intensity mental-health intervention Aimed at individual adult women facing adversity Have 5 weekly 90 min sessions administered in a particular order Can be facilitated by lay workers Provides handouts and activities to be used between sessions	Adapted for out-of-school adolescents Adapted to be offered in group settings to promote mental wellness and prevent mental illness Incorporated age-appropriate activities for practice between sessions (e.g., Session 3 included playing football, dancing, etc., as behavior-activation activities and Session 1 included thinking about preparing different locally available fruits as a means of managing stress) Group session duration–3 h Groups with up to 8 participants or more The manual included specific case examples for youth mentors to practice integrating the different curricula Change in images and language to contextualize for target population Individualized PM+ adapted to be offered for 1 h weekly sessions by youth mentor

**Table 2 ijerph-21-00223-t002:** Overall bridging the gaps training schedule.* REPACTED—Nakuru Site Community Led Organization, the Rapid Effective Participatory Action in Community Theatre Education and Development, located in Rhonda; * IECE—Nairobi Site Community Led Organization, the Integrated Education for Community Empowerment, located in Kariobangi.

Day	Content	Activities	Teams
Day 1	Introduction to teams Logistics Understanding the Feasibility Study Understanding Mentorship and Facilitation	Sharing documents and contacts Lecture sessions Practice sessions	Population Council REPACTED * IECE * Depending on the site
Day 2 and 3	Pre-assessment Safety Planning Gender and Gender Norms Curriculum Financial Education Curriculum Reproductive Health Curriculum Life Skills Curriculum	Lectures Group Work practical	Population Council REPACTED IECE Depending on the site
Day 4 to Day 13	Introduction to mental health and contextualizing it to the target population Role of mentorship in mental health promotion, prevention, and management Common Mental Disorders among adolescents Identifying and reporting mental ill-health to caregivers Task sharing in mental health WHO PM+ Mapping referral pathways Mental-health supervision Helper self-care for peer support Integrating all curricula	Lectures Group work Individual reflection Role-plays Group work practical	Mental-Health Team
Days 14 and 15	Review of Day 2 and 3 content More practice in integrating the curricula ODK training and practice for participant recruitment	Practice sessions	Population Council REPACTED * IECE * Depending on the site

**Table 3 ijerph-21-00223-t003:** Example PM+ specific Supervision Report July-August 2021.

Safe Space Supervisory Report (Prevention, Promotion, and Management of Mental Health)			
SITE	GROUP PM+	INDIVIDUAL PM+ (High-Risk Cases)	REFERRALS
Rhonda	19 participants had completed the 5 PM+ safe space sessions	0	2 made as follows: 1-Suicide risk–psychological issues caused by financial stress 1-Financial Issues–Technical training for livelihood
	85 participants still had not gone through all the 5 weeks of PM+ sessions	3 participants were enrolled and sessions were ongoing	1-Intimate Partner Violence –referred to a rescue home but later relocated to the village 1 adolescent needed to be referred for psychological care as suicidality was likely, based on the Study Team assessment. The youth mentor informed.
Kariobangi	78 participants had completed all sessions	0	9 referrals were made as follows: 3-antenatal care 2-SGBV care to report and hospital for physical and psychological well-being 2-psychological counseling at Health Centre 1 for economic empowerment at a local organization 1 for legal documentation as they were a foreigner
	54 participants still had not gone through the 5 weeks of the curriculum	7 participants were enrolled and sessions were ongoing	

## Data Availability

Data are contained within the article and Appendix A.

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
