# Peer review of "Integrating Mental Health Management into Empowerment Group Sessions for Out-of-School Adolescents in Kenyan Informal Settlements: A Process Paper"

_ijerph, 2024, doi:10.3390/ijerph21020223_

Round 1

Reviewer 1 Report

Comments and Suggestions for Authors

The authors present an interesting topic: Bridging the Gaps: Integrating mental health management into youth led health and life skills group mentorship sessions, for out of school adolescents in urban informal settlements.

The target population is vulnerable and it is important that mental health is addressed using innovative ways as conducted by the research team.

While the manuscript details the process involved in bridging the gaps the authors indicate that this was reflections on the process which does not tally with what is reported. Indeed there is no section on reflections and there is no need to have one as no data was collected on reflections by the study team.

Introduction: This was interesting and well written.

Methods: What was the rationale of recruiting 477 instead of the targeted 400 participants? Was it  this ethical? Normally a 10% loss to follow up is adequate.

The consort flow diagram as well as the text o page 4 should indicate the time frame of the follow up time points in weeks or months.

How were the homes of out of school adolescents identified from which the participants were recruited?

How was the adaptation of the curriculum in line 266 and 274 done, please elaborate. 

In section 2.3.4; how were the recommendations captured and how were they graded or scored by the mentors to be considered for involvement in the implementation process?

Comments on the Quality of English Language

The grammar is well written however there is need for conciseness and avoidance of repetition.  The flow of events is also a mishmash making it difficult for a smoother read. For example the formative research processes are presented after the procedures on page 3.

On page 6 section 2.3.2 is a section on how the study began, this is totally  misplaced.

The introduction section 3rd sentences has 6 lines. This could be shortened.

Line 48 has a hanging statement on urban informal........

Author Response

Dear Reviewer 1, 

Thank you for taking your time and sharing your insights on our paper. Kindly find attached our responses to your specific points. 

Best Regards,

Joan. 

Reviewer 2 Report

Comments and Suggestions for Authors

Overall:

This is an article that addresses an important topic. It targets an underserved population with urgent needs for socioemotional education and support. The large sample size in a low resource setting gives it high potential for future replications and scaling up. I especially appreciate the author’s commitment to social equity in the midst of a pandemic.

There are some concerning major issues for revision:  1) lack of process or health outcome measurements (either qualitative or quantitative), 2) low clarity in the abstract, 3) redundancy of language in several parts, 4) missing “results section”, and 5) a weak discussion without sufficient references to support using this one approach, comparable studies in other low-resource settings, or alternative approaches to the existing problems.

Specific comments:

Title

Long and hard to read. Think about ways to shorten it to at most 50% of the word count and highlight only what’s essential in the title. Readers can find specifics in the abstract and the manuscript.

Abstract

Appreciate the difficulty of summarizing a complex study. There is a need for significant editing in language. Please refer to this article for abstract writing tips.

https://www.bmj.com/content/346/bmj.f2974

Sentences are too long and hard to read. For example, the first sentence, the subject and the verb are separated by 3 lines using passive tense.

Please specify the time of the study (other than alluding the pandemic, which is at least 3 years long), sample size, qualitative vs. quantitative analysis, and conclusion.

Introduction

Line 47: negative adversities=adversities. Adversities, by definition, are negative.

References: try to un-group them by inserting them into specific parts of the sentence that can be backed up by each article, instead of attaching them in a row at the end of a sentence with multiple parts.

The aim is stated here. Please insert this into the abstract above the main text too. What are some pre-implementation hypotheses regarding whether an intervention like this may not may not have measurable benefits in this setting? How would we know that the program is feasible, implemented as intended or successful?

Methods

Please specify your power calculations.

How was 15-18 years-old chosen? Adolescence is generally considered to start at 10-12.

Was PHQ-9 culturally adapted and validated in Kenya prior to this study?

Please reference publications verifying that girls are “more likely to experience adverse outcomes” and be specific about “adverse outcomes”. There is evidence that boys and girls have different negative outcomes—boys tend to externalize e.g. bullying, fights, and girls tend to internalize e.g. depression, anxiety. Was it harder to recruit boys or the recruiters intentionally skipped potential male candidates?

Please be careful with the use of “N” vs “n”. “N” generally referred to the total sample. “n” can be used for subsamples.

Figure 1: please spell out BTG, and this goes to all figures and table titles.

Line 188: please highlight a few more key details regarding the mentor training crucial in implementation and replication, e.g. length, incentives, or at least mention where in the article to find more details. Readers may not be reading the entire methods section at a first glance.

Line 209: This is an interesting and important part of implementation. Please provide the questions asked during formative research in an appendix. Were the interviews recorded? How were the conclusions drawn—e.g. formal qualitative analysis? Manual coding?

Figure 3: the right column seems to lack an edge

I am curious why “contraceptives”, “teenage pregnancy” and “romantic relationships” were not taught to boys. They seem important for both genders, though may need slightly different focus.

Please format the figure so that life skill sessions and financial education sessions for each gender would be at the same visual level.

It seems like boys received fewer sessions than boys in financial education. Is there any evidence behind this? Similarly, I am a bit surprised that girls received many fewer sessions on gender and gender norms. What is the evidence behind it, if any? Were the two genders separated in these teaching sessions based on local norms? Otherwise, I think it would actually be a good idea to promote open discussion around some of these important topics.

Lines 363-373: were there any mentors who didn’t meet the criteria of qualification?

Results: Did the authors skip the results section? What are the measurements of success? What are the process indicators of fidelity? Can you summarize the main lessons learned?

Discussion

Please clearly state the strengths and limitations.

How do your findings compare to other low-middle-income countries or the world at large? This could be important to state for readers to understand and to apply your findings elsewhere. There is almost no reference under “discussion”.

There needs to be more in-depth discussion about what success would look like for future studies and what other programs exist in literature for improving outcomes in vulnerable youth similar to this study’s population.

Conclusion

Could be shortened and more focused on recommendation. Lines 539-546 are very much descriptive content repetition from above.

Comments on the Quality of English Language

Needs significant revision in language and organizing ideas. This is a good article for general principles to follow throughout the article:

https://www.nature.com/articles/d41586-019-00546-7

Author Response

Dear Reviewer 2, 

We are grateful for your thoughts and insights on our paper. Kindly find attached our responses. We hope we have adequately responded and are having to make additional changes that you might have. 

Regards,

Joan. 

Reviewer 3 Report

Comments and Suggestions for Authors

This study describes a training session developed for out of school adolescents in Kenyan urban informal settlements. I have several comments:

1.      The abstract should be better structured and focus more on the development of the training session rather than the background.

2.      The introduction part needs to be boosted especially about research aims.

3.      The structure of this paper is not easy to follow. At the end of the introduction, the authors need to describe what will be included in the following sections.

4.      The section Materials and Methods needs to be structured better. Research backgrounds, research methods and some results were mixed in this chapter. It would be difficult for the readers to follow.

5.      This process paper is not easy to follow though I could see the importance of the study. However. the whole story needs to be described more concisely and in a more organised way. 

Comments on the Quality of English Language

Some sentences were too long and too complicated. 

Author Response

Dear Reviewer 3, 

Thank you so much for accepting to review our paper. Attached are our responses to your comments and we hope we have adequately responded to your concerns. 

Many thanks!

Joan. 

Round 2

Reviewer 3 Report

Comments and Suggestions for Authors

This paper has improved a lot and is suitable for publication.

Comments on the Quality of English Language

none

Author Response

Dear Reviewer 3, 

Thank you for your comments on the manuscript this second round. We have done a comprehensive review of English language use and grammar. 

Sincerely, 

Joan. 
